# Deep Identification of Propagation Trees in Graph Diffusion

## Abstract

Understanding how information or influence propagates through a network—such as during an epidemic outbreak or the spread of misinformation—is a fundamental yet challenging problem. While prior works have focused on cascade prediction (forecasting future infected nodes), network inference (recovering latent global diffusion graphs), or source localization (identifying diffusion's origin), these approaches do not recover the actual "who-infected-whom" propagation tree for a specific diffusion instance. We introduce DIPT (Deep Identification of Propagation Trees), a probabilistic framework that infers propagation trees from final observed node diffusion states, without knowledge of the diffusion mechanism. DIPT models local influence strengths between nodes and uses a discrete-continuous alternating optimization strategy to jointly learn the diffusion mechanism and infer the propagation structure. Empirical results across eight real-world datasets demonstrate that DIPT consistently outperforms existing approaches in reconstructing propagation trees.

## 1 Introduction

Graph inverse problems aim to uncover the underlying causes of observed phenomena in networks (Wang et al., 2022; Ling et al., 2022). A prominent example is *diffusion source localization*, which identifies the origin(s) of spread from the final infection state. Recent GNN-based models have advanced this task (Ling et al., 2022; Yan et al., 2024; Wang et al., 2023; Ling et al., 2024). However, source localization alone provides limited insight into the underlying diffusion dynamics. As shown in Fig. 1(a), source localization identifies only the origin nodes among the infected(pink). Hence, to capture the full transmission pathways—revealing not just the sources, but also the sequence of infections and who influenced whom as shown in Fig. 1(b), without knowing the underlying diffusion mechanism is a challenging problem. This paper focuses on such a problem which we call propagation tree identification.

Understanding diffusion pathways is crucial across domains. In infectious disease modeling, identifying who infected whom—known as *contact tracing*(Mokbel et al., 2020; Kleinman & Merkel, 2020)—enables interventions to prevent further spread from initial carriers. In misinformation campaigns, tracing how false content propagates helps to mitigate network-wide spread(Zhou & Zafarani, 2019). In phylogenetics, mapping genetic mutations reveals evolutionary lineages (Penny, 2004; Bouckaert et al., 2014). In all cases, transmission typically begins from a few sources and spreads in tree or forest-like structures, where each infected node may influence multiple others.

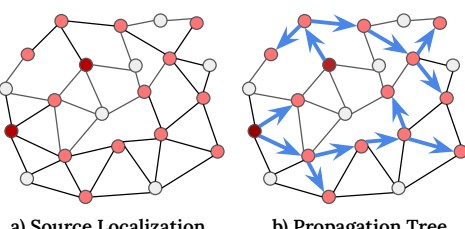

a) Source Localization    b) Propagation Tree Identification

Figure 1: Given the diffused state (pink color nodes), source localization aims to identify the source nodes (red) only (a), while propagation trees identification can further reveal how infection spreads, as shown in blue arrows (b).

There are research areas and works related yet distinct from propagation tree identification. The first type is prior-knowledge-based, such as Steiner-tree methods, which require the diffusion mechanism, edge costs, or influence scores (Mishra et al.,

2023). These methods typically optimize a maximum-likelihood objective over possible trees but lack learnable components, limiting their adaptability to real-world settings. In practice, such heuristics are often unavailable in domains like disease spread, e-commerce behavior, or rumor propagation. A separate line of work focuses on inferring global influence graphs from timestamped cascade data (Gomez-Rodriguez et al., 2012). While these methods are learning-based, they operate on multiple cascades and infer global diffusion networks. In contrast, a third class of learning-based methods—closer to our setting—models cascade dynamics from final infection states. However, they typically aim to predict node states at fixed time points, without recovering the underlying "who-infected-whom" paths essential to propagation tree identification (Čutura et al., 2021; Yan et al., 2024). Consequently, learning these propagation paths without assuming a specific diffusion mechanism remains underexplored due to following nontrivial challenges: *1) Difficulty of inferring the unknown propagation mechanism.* Existing methods often rely on known diffusion models such as Linear Threshold (LT), Independent Cascade (IC), which make strong and often unrealistic assumptions about how information spreads. These models struggle to capture the complexity of real-world diffusion, where infection dynamics also depend on node attributes—e.g., a rumor's transmission from user A to B might depend on their trust level, interest alignment, or social context. *2) Intractable search space.* The number of valid propagation trees leading to the observed infected nodes grows exponentially with the size of the network, making joint inference over source nodes and propagation edges highly intractable. *3) No or incomplete observation of propagation tree during training.* In realistic scenarios, only a partial observation of the diffusion process at edge level is available. For example, in epidemiological modeling, contact tracing rarely provides the full transmission path, making it challenging to learn from partial or no observations during training (Rodriguez et al., 2014).

To address these challenges, we propose a probabilistic framework called **DIPT** (Deep Identification of Propagation Trees), which infers latent propagation trees from final infection observations. To tackle *Challenge 1* (unknown propagation mechanisms), DIPT models markov-structured diffusion recursively: each node's infection probability is conditioned on its parent's state and their node-level features, enabling the model to learn influence without relying on rigid assumptions. For *Challenge 2* (exponential search space), we constrain inference by learning a prior over source nodes and inferring the propagation tree that maximizes the likelihood of the observed infections. To overcome *Challenge 3*, we employ a discrete-continuous alternating optimization strategy that jointly learns the propagation tree and the diffusion model, without need for edge-level annotations. Our main contributions are:

• We propose a graph inverse problem called propagation tree identification, which aims to infer "who infected whom" given the final diffusion state, without assuming the underlying diffusion mechanism.

• We formulate a mathematical problem for the propagation tree identification which is a maximum-likelihood of a Bayesian network characterizing the diffusion process.

• We develop a discrete-continuous *alternating optimization framework* to jointly infer propagation trees(discrete problem) and learn diffusion process(continuous problem) without access to ground truth paths.

## 2 PREVIOUS WORK

**Propagation Path Reconstruction**. Early methods estimate global edge weights from historical interactions and use heuristics like LeaderRank with probabilistic inference to recover global directed graphs (Gomez-Rodriguez et al., 2012; Zhu et al., 2016). Deep models such as I3T and DeepIS (Tai et al., 2023; Xia et al., 2021) combine GNNs, sequence encoders (e.g., Bi-LSTM), and attention to capture structural and temporal diffusion features. Others improve scalability through community-aware path modeling (Zhu et al., 2023), though these methods focus on forward diffusion from known sources. A separate line of direction frames path reconstruction as a Steiner tree problem, recovering cascades via minimal-cost trees over observed infections (Mishra et al., 2023; Jang et al., 2021; Rozenshtein et al., 2016). While effective, these methods depend on heuristics like hand-crafted cost functions, assumptions about edge weights or infection timestamps, limiting their adaptability in realistic settings. Recent generative approaches instead aim to infer diffusion paths from final observed node states. Qiu et al. (2023) reconstructs diffusion histories from a single

final snapshot by estimating posterior barycenters with MCMC and GNN-guided proposals. Čutura et al. (2021) use a Variational Autoencoder to model node trajectories, but performance degrades as time horizons grow. DDMSL (Yan et al., 2024) employs discrete diffusion models for source localization and snapshot prediction, yet both fail to reconstruct explicit transmission edges—i.e., the "who-infected-whom" links critical for full path recovery.

**Source localization**. Classical approaches estimate one or more sources under SI/SIR models using full or partial observations (Prakash et al., 2012; Wang et al., 2017; Zhu et al., 2016; Zhu & Ying, 2016; Zang et al., 2015; Zhu et al., 2017). GNN-based models (Dong et al., 2019) improved accuracy, but often lack uncertainty quantification and incur high inference costs on large graphs. More recent methods (Ling et al., 2022; Wang et al., 2022; 2023; Xu et al., 2024) learn latent source distributions without strict diffusion assumptions. However, these are limited to localizing sources or reconstructing discrete-time infection states, without recovering full propagation edges.

## 3 DEEP IDENTIFICATION OF PROPAGATION TREES

### 3.1 PROBLEM FORMULATION

Given a graph $G = (V, E, \mathbf{y})$, where $V$ is the set of nodes, $E \subseteq V \times V$ is the set of edges, and $\mathbf{y} \in \{0, 1\}^{|V|}$ is a binary infection state vector indicating whether node $v$ is infected ($\mathbf{y}_v = 1$) or not ($\mathbf{y}_v = 0$), the objective is to reconstruct a propagation tree $\mathcal{T}$ given the observed infections. Information spreads through directed propagation trees rooted at source nodes $\mathbf{s} \in \{0, 1\}^{|V|}$, where $\mathbf{s}_v = 1$ denotes that node $v$ is a source. Each node $v \in V$ is associated with a feature vector $\mathbf{F}_v \in \mathbb{R}^d$, forming a node feature matrix $\mathbf{F} \in \mathbb{R}^{|V| \times d}$ where d is the feature dimension.

### 3.2 OBJECTIVE FUNCTION

We formulate propagation tree identification as a maximum a posteriori (MAP) estimation problem. The goal is to jointly infer the source distribution $\tilde{s}$ and propagation tree $\tilde{\mathcal{T}}$ that maximize the joint probability $P(s, \mathbf{y} \mid \mathcal{T}, G)$ of observed infections:

$$\tilde{s}, \tilde{\mathcal{T}} = \arg\max_{s, \mathcal{T}} P(s, \mathbf{y} \mid \mathcal{T}, G) \tag{1}$$

However, equation 1 cannot be solved directly. Since the infection state $\mathbf{y}$ is influenced by the graph topology $G$, source nodes, $s$, and the propagation tree $\mathcal{T}$, we can decompose the problem. This reformulation simplifies the Maximum A Posteriori (MAP) estimation as follows:

$$P(s, \mathbf{y} \mid \mathcal{T}, G) = P(\mathbf{y} \mid s, \mathcal{T}, G) \cdot P(s) \tag{2}$$

where $P(s)$ represents prior distribution over source nodes and $P(\mathbf{y} \mid s, \mathcal{T}, G)$ represents likelihood of observed infections given the source nodes and propagation tree. We model the likelihood and prior using parameterized functions: $P_\psi(\mathbf{y} \mid s, \mathcal{T}, G)$ captures the diffusion process by learning edge-level influence, and $P_\phi(s)$ encodes a learnable prior over source node distributions. This leads to the following training objective:

$$\max_{\mathcal{T}, \phi, \psi} P_\psi(\mathbf{y} \mid s, \mathcal{T}, G) \cdot P_\phi(s) \tag{3}$$

As $\mathcal{T}$ is unobserved and intractable, we optimize the objective in Eq. 3 by alternately updating (1) the propagation structure $\mathcal{T}$ given current model parameters (Section 3.4), and (2) the model parameters $(\phi, \psi)$ given the inferred propagation tree. This alternating optimization is summarized in Algorithm 1 and its convergence is discussed in **Theorem 2** in Appendix D.2.

### 3.3 ESTIMATING THE DIFFUSION PROCESS

We model the likelihood term $P_\psi(\mathbf{y} \mid s, \mathcal{T}, G)$ in equation 3 using a recursive diffusion process along the inferred propagation tree $\mathcal{T}$. The joint infection probability is factorized as a product of local conditionals over the tree structure:

$$P_\psi(\mathbf{y} \mid s, \mathcal{T}, G) = \prod_{v \in V \setminus s} P_\psi(y_v \mid y_{\mathrm{Pa}(v, \mathcal{T})}), \tag{4}$$

where $V \setminus s$ denotes the set of non-source nodes, and $\mathrm{Pa}(v, \mathcal{T})$ denotes the parent of node $v$ in the tree $\mathcal{T}$. This factorization imposes a tree-structured Markov property in which each node is

---

**Algorithm 1** DIPT Training via **Alternating Optimization**

---

**Input:** $G = (V, E)$, features $F$, infections $\mathbf{y}$, steps $K$, epochs $E$, weights $\lambda, \mu$
**Output:** $\psi, \phi$
Init $\psi, \phi$; **for** $e = 1$ **to** $E$ **do**
$\quad I_{uv} = f_\psi(F_u, F_v), \; \forall (u, v) \in E$ ;           // equation 5
$\quad$ Sample $s \sim P_\phi(s)$ ;           // Sec. 3.5
$\quad P_{\text{inf}}^{(0)}(v) = P_\phi(s_v)$; **for** $k = 1$ **to** $K$ **do**
$\quad\quad P_{\text{inf}}^{(k)}(v) = \max\big(P_{\text{inf}}^{(k-1)}(v), \; \max_{u \in \mathcal{C}(v)} P_{\text{inf}}^{(k-1)}(u) I_{uv}\big), \forall v$
$\quad$ **end**
$\quad p_v = \arg\max_{u \in \mathcal{C}(v)} P_{\text{inf}}^{(K)}(u) I_{uv}$ for $y_v = 1, v \notin s$; $\mathcal{T}^* = \{(p_v, v)\}$ ;     // (10)
$\quad P_\psi(\mathbf{y} \mid s, \mathcal{T}^*, G) = \prod_{v \notin s} P_\psi(y_v \mid y_{p_v})$ ;     // equation 4
$\quad$ Compute $\mathcal{L}_{\text{diff}}, \mathcal{L}_{\text{ELBO}}$; if $\mathcal{T}_{\text{obs}}$: $\mathcal{L}_{\text{obs}} = -\sum_{(u,v) \in \mathcal{T}_{\text{obs}}} \log P_\psi(y_v \mid y_u)$ ;     // equation 13
$\quad \mathcal{L} = \lambda \mathcal{L}_{\text{diff}} + \mathcal{L}_{\text{ELBO}} + \mu \mathcal{L}_{\text{obs}}$ ;     // equation 14
$\quad$ Update $\psi, \phi$ via $\nabla \mathcal{L}$
**end**
**return** $\psi, \phi$

---

conditionally dependent only on its parent. Each local conditional is parameterized by a learnable *edge transmission* produced from node features:

$$m_\psi\big(\text{Pa}(v) \to v\big) := f_\psi\big(\mathbf{F}_v, \mathbf{F}_{\text{Pa}(v)}\big) \in (0, 1), \tag{5}$$

where $\mathbf{F}_v$ and $\mathbf{F}_{\text{Pa}(v)}$ denote the feature vectors of node $v$ and its parent, respectively, and $m_\psi(\text{Pa}(v) \to v)$ is the edge transmission probability. The marginal infection probability of a child depends recursively on the marginal of its parent:

$$P_\psi\big(y_v = 1 \mid s, \mathcal{T}, G\big) = P_\psi\big(y_{\text{Pa}(v)} = 1 \mid s, \mathcal{T}, G\big) \, m_\psi\big(\text{Pa}(v) \to v\big), \quad \text{with } P_\psi(y_u = 1 \mid s, \mathcal{T}, G) = 1 \text{ for } u \in s.$$

To learn the diffusion process, we optimize the following negative log-likelihood over observed infection states:

$$\mathcal{L}_{\text{diff}} = -\sum_{v \in V \setminus s} \Big[ y_v \log\Big( P_\psi\big(y_{\text{Pa}(v)} \mid s, \mathcal{T}, G\big) \, m_\psi\big(\text{Pa}(v) \to v\big)\Big)$$
$$+ \, (1 - y_v) \log\Big(1 - P_\psi\big(y_{\text{Pa}(v)} \mid s, \mathcal{T}, G\big) \, m_\psi\big(\text{Pa}(v) \to v\big)\Big)\Big]. \tag{6}$$

This loss enables the model to learn localized influence patterns across the tree structure and generalize the underlying diffusion dynamics from observed infection outcomes.

### 3.4 OPTIMIZING PROPAGATION TREES

Since the true propagation tree is unobserved during training, we estimate $\mathcal{T}^*$ by maximizing equation 3 while keeping the model parameters fixed with respect to $\mathcal{T}$. The inferred tree consists of directed edges linking seed nodes $s$ to observed infected nodes $\mathbf{y}$:

$$\mathcal{T}^* = \arg\max_{\mathcal{T}} \; P_\psi(\mathbf{y} \mid s, \mathcal{T}, G) \cdot P_\phi(s). \tag{7}$$

To approximate this structure, we first compute edge-level influence scores using $f_\psi$ for all edges in $G$ (equation 5), forming a sparse influence matrix[1] with entries $\mathbf{I}_{uv} = f_\psi(\mathbf{F}_u, \mathbf{F}_v) = m_\psi(u \to v)$, where each entry represents the learned transmission strength along edge $(u, v)$.

Propagation scores are then updated iteratively in a monotonic increasing way. At iteration $k = 0$, initial infection probabilities are given by the source prior:

$$P_{\text{inf}}^{(0)}(v) = P_\phi(s_v). \tag{8}$$

At iteration $k \geq 1$, the probability of node $v$ being infected is updated by

$$P_{\text{inf}}^{(k)}(v) = \max\left( P_{\text{inf}}^{(k-1)}(v), \; \max_{u \in \mathcal{C}(v)} P_{\text{inf}}^{(k-1)}(u) \, \mathbf{I}_{uv} \right), \tag{9}$$

---

[1]We do not store a full influence matrix $\mathbf{I} \in \mathbb{R}^{|V| \times |V|}$; influence scores are computed only for existing edges $(u, v) \in E$.

where $\mathcal{C}(v)$ denotes the set of candidate parents of $v$, i.e. its infected neighbours in $G$. Because improvements can only be inherited from nodes that improved in the previous iteration, this induces a topological order on updates. Consequently, the set of parent selections defines a directed diffusion tree. Monotonicity and acyclicity proof are discussed in Theorem 1 (Appendix D.1,D.4). After $K$ iterations, the propagation tree $T^*$ is constructed by assigning to each infected node $v$ the parent that maximizes its inherited likelihood under the monotone update from neighbors that became infected in earlier iterations[2],

$$p_v = \arg \max_{u \in \mathcal{C}(v) \,:\, t(u) < t(v)} P_\psi(y_u)\, m_\psi(u \to v), \tag{10}$$

which, by construction, respects the topological order induced by the updates and yields an acyclic propagation tree.

## 3.5 LEARNING THE PRIOR OF SEED NODES

The intrinsic patterns of the prior over seed nodes, $P(s)$, are hard to model and often high dimensional, which leads to intractability. To tackle it, we map $s$ to a latent embedding $z$ in lower dimensional space representing the abstract semantics. A variational inference framework is used to learn an approximation of $P(s)$, capturing its structure and variability. It consists of a generative process $P_{\phi_D}(s \mid z)$[3] that reconstructs $s$ from $z$, and a simple prior $P(z)$, typically a standard Gaussian $\mathcal{N}(0, I)$, which regularizes the latent space. The variational posterior $q_{\phi_E}(z \mid s)$ approximates the intractable $P(z \mid s)$ and serves as the encoder. Variational inference is introduced to efficiently approximate the intractable posterior $P(z \mid s)$ by optimizing the Evidence Lower Bound (ELBO), ensuring that the latent variables $z$ capture the variability of $s$ while maintaining regularization through the prior $P(z)$. The objective is to maximize the Evidence Lower Bound for $P(s)$:

$$L_{\text{ELBO}} = \mathbb{E}_{q_{\phi_E}(z|s)} \left[\log P_{\phi_D}(s \mid z)\right] - \text{KL}\left(q_{\phi_E}(z \mid s) \| P(z)\right). \tag{11}$$

The first term ensures accurate reconstruction of $s$, while the second term regularizes the latent distribution to match the prior $P(z)$. Learning prior over source nodes directly addresses the challenge of intractable search space (Challenge 2). In real-world diffusion, source nodes are often sparse, learning a prior provides a compact representation of source configurations, enabling the model to learn a distribution over plausible sparse source sets. Sampling from this learned prior avoids the need to enumerate an exponential number of candidate sources, thereby narrowing the joint search space over source nodes and propagation trees.

## 3.6 INCORPORATING PARTIAL PROPAGATION PATH OBSERVATIONS

The propagation tree is unobservable during training in our problem setting, so the diffusion process is inferred solely from the observed infection states $\mathbf{y}$. However, in some scenarios when partial observations of the propagation tree $\mathcal{T}_{\text{obs}} \subset \mathcal{T}$ are available, DIPT can also incorporate such supervision to improve training. In such setting, the objective function in Eq. 1 is extended to:

$$P(s, \mathbf{y} \mid \mathcal{T}_{\text{obs}}, \mathcal{T}_{\text{unobs}}, G) = P(\mathbf{y} \mid s, \mathcal{T}_{\text{obs}} \cup \mathcal{T}_{\text{unobs}}, G) \cdot P(\mathcal{T}_{\text{obs}} \mid s, \mathbf{y}, G) \cdot P(s) \tag{12}$$

The unobserved portion $\mathcal{T}_{\text{unobs}}$ is inferred as before, while observed edges $(u, v) \in \mathcal{T}_{\text{obs}}$ provide a supervised loss signal and maximize the conditional probability $P_\psi(\mathbf{y}_u \mid \mathbf{y}_v)$:

$$\mathcal{L}_{\text{obs}} = -\sum_{(u,v) \in \mathcal{T}_{\text{obs}}} \log P_\psi(\mathbf{y}_v \mid \mathbf{y}_u) \tag{13}$$

This objective encourages the model to assign higher likelihood to edges consistent with known propgation tree edges using direct supervision. When supervision is available, the training loss becomes:

$$\mathcal{L}_{\text{total}} = \mathcal{L}_{\text{ELBO}} + \lambda \cdot \mathcal{L}_{\text{diff}} + \mu \cdot \mathcal{L}_{\text{obs}} \tag{14}$$

where $\lambda$ and $\mu$ are hyperparameters. This formulation enables DIPT to transition smoothly between unsupervised and partially supervised training.

---

[2]Here $t(v)$ denotes the iteration index at which node $v$ first becomes infected, i.e., the first iteration where $P_{\text{inf}}^{(k)}(v)$ strictly increases.

[3]$\phi_D$ and $\phi_E$ represents decoder and encoder respectively

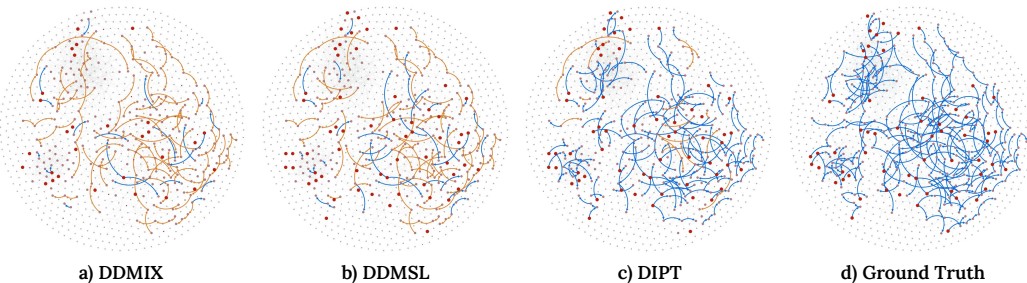

a) DDMIX      b) DDMSL      c) DIPT      d) Ground Truth

Figure 2: Predicted vs. ground truth propagation trees on Memetracker. Correctly predicted edges are shown in blue, incorrect ones in orange. Source nodes are marked in red, and infected nodes in pink.

### 3.7 TEST TIME INFERENCE OF PROPAGATION TREE

At inference time, we aim to infer the optimal propagation tree given an observed infection state $\mathbf{y}$. A learnable latent vector $\hat{z}$ is initialized, which is optimized while freezing model parameters $\phi$ and $\psi$. This enables test-time optimization of $P_\phi(s)$ through $z$ rather than sampling directly. The latent vector $\hat{z}$ is initialized as the empirical average $\bar{z}$ over the training set and optimized to maximize the expected likelihood of observed infections:

$$\mathcal{L}_{\text{inf}} = \max_{\hat{z}} \mathbb{E}_{s \sim P_{\phi_D}(s|\hat{z})} \left[ P_\psi(\mathbf{y} \mid s, \mathcal{T}^*, G) \right] - \gamma \cdot \|\hat{z} - \bar{z}\|^2 \tag{15}$$

The regularization term constrains $\hat{z}$ to remain close to the training distribution, mitigating test-time drift. The propagation tree $\mathcal{T}^*$ is inferred from $s \sim P_{\phi_D}(s \mid \hat{z})$ using the same inferring propagation tree inference process described in training(see Eq. (8-11)). The constrained objective function Eq. equation 15 cannot be computed directly, so we provide a practical version of the inference objective function: since the diffused observation $y$ fits the Gaussian distribution and the seed set $s$ fits the Bernoulli distribution(check Appendix B.1), we can simplify Eq. (16) as:

$$\mathcal{L}_{\text{inf}}^* = \min_{\hat{z}} \left[ -\sum_{i=1}^{N} \log \left( f_{\phi_D}(\hat{z})^{s_i} \cdot (1 - f_{\phi_D}(\hat{z}))^{1-s_i} \right) \quad + \frac{1}{2} \|\hat{\mathbf{y}} - \mathbf{y}\|^2 + \lambda \|\hat{z} - \bar{z}\|^2 \right]$$

Here, the first term captures the Bernoulli likelihood of the seed nodes, and the second term represents the Gaussian likelihood. The test-time inference process does not introduce any new training objective. Instead, it optimizes the latent source representation $\hat{z}$ given the observed diffused state $\mathbf{y}$, while keeping the learned diffusion model fixed. This enables instance-specific adaptation through the same likelihood computation and propagation tree inference steps used during training (see Algorithm 2 in Appendix B), and a **theoretical analysis** of the error correction of propagation tree is provided in Appendix D.3.

## 4 EXPERIMENTAL EVALUATION

### 4.1 DATASETS AND EVALUATION METRICS

We evaluate DIPT and baselines on eight real-world graph datasets. Memetracker (Rodriguez et al., 2011) captures real diffusion via hyperlink cascades across online articles. We use a subnetwork with 583 sites and 6.7k cascades, extracting source nodes as the earliest 5% per cascade and each cascade is converted into a propagation tree. Cora-ML (Rossi & Ahmed, 2015), CiteSeer (Wu et al., 2016), and Power Grid (Watts & Strogatz, 1998) provide the node features and topology of the citation networks and infrastructure grids respectively. Cascades are obtained by simulating diffusion by randomly selecting 10% of nodes as sources and running the SI model for 200 steps until convergence. Twitter and Weibo are large-scale real-world datasets of information diffusion in social networks, with 616k and 490k nodes respectively, where each cascade records user-to-user retweet

Table 1: Performance comparison for Propagation Tree Identification. Best results are in bold. Prec. = Path Precision, Jac. = Jaccard Index.

| Method | Cora-ML | | Memetracker | | CiteSeer | | Power Grid | | IDSS | | Twitter | | Weibo | | Pol | |
|---|---|---|---|---|---|---|---|---|---|---|---|---|---|---|---|---|
| | Prec. | Jac. | Prec. | Jac. | Prec. | Jac. | Prec. | Jac. | Prec. | Jac. | Prec. | Jac. | Prec. | Jac. | Prec. | Jac. |
| DDMIX | 0.327 | 0.195 | 0.062 | 0.041 | 0.236 | 0.133 | 0.081 | 0.031 | 0.109 | 0.057 | 0.211 | 0.122 | 0.203 | 0.118 | 0.198 | 0.115 |
| DITTO | 0.371 | 0.218 | 0.091 | 0.052 | 0.312 | 0.178 | 0.106 | 0.050 | 0.115 | 0.061 | 0.258 | 0.154 | 0.210 | 0.124 | 0.242 | 0.148 |
| DDMSL | 0.412 | 0.259 | 0.119 | 0.063 | 0.405 | 0.253 | 0.130 | 0.069 | 0.121 | 0.064 | 0.312 | 0.187 | 0.216 | 0.130 | 0.298 | 0.176 |
| DIPT | **0.622** | **0.452** | **0.602** | **0.430** | **0.593** | **0.421** | **0.680** | **0.515** | **0.421** | **0.266** | **0.581** | **0.418** | **0.552** | **0.397** | **0.565** | **0.404** |

paths over time. For our problem setting, we extract the final diffusion state and treat the observed retweet tree as ground truth. Similarly, Pol (Conover et al., 2011) is a temporal retweet network about a U.S. political event. Further dataset preprocessing details appear in Appendix B.3.

**Infectious Disease Simulated Spread (IDSS) Dataset.** To evaluate performance in large-scale, realistic epidemics, we simulate disease spread using a spatial SIR model (Kermack & McKendrick, 1932) over 3,143 U.S. counties, incorporating real mobility data (Kang et al., 2020; SafeGraph, Inc.). Infections seed via air travel and evolve under a time-varying reproduction number. Each run generates a propagation forest over 500–1,000 counties and 12k–48k infections. Full details and data are in Appendix A.

**Evaluation Metrics.** We evaluate DIPT on two tasks: propagation tree identification and source localization. For tree identification, we use *Jaccard Index* and *Path Precision*. Jaccard captures structural overlap between predicted and ground-truth edges, while Path Precision reflects correctness along true diffusion paths. For source localization, we use standard classification metrics to asses accuracy of source localization.

### 4.2 COMPARISON METHODS AND EXPERIMENTAL SETTINGS

Most prior approaches focus on predicting global diffusion graphs from cascades, using *non-learnable heuristics*, MLP-like models, or classical methods that assume timestamped infections and fixed diffusion mechanisms. These differ fundamentally from DIPT, which reconstructs instance-level propagation trees from a single final infection state. We therefore compare against learnable diffusion models which can capture infection dynamics from final diffusion observations, as:

- *Generative Diffusion Models.* **DITTO** (Qiu et al., 2023) reconstructs diffusion histories from a single final snapshot by estimating posterior barycenters with MCMC and GNN-guided proposals. **DDMIX** (Čutura et al., 2021) uses a VAE to reconstruct diffusion trajectories by learning latent node states. **DDMSL** (Yan et al., 2024) models invertible diffusion to recover node-level timestamps. These methods are adapted to approximate propagation paths by combining random forward and backward walks(MCMC in case of DITTO) between nodes active in adjacent diffusion steps. This enables a fair comparison to DIPT in terms of reconstructing propagation tree.

- *Source Localization Methods.* We also benchmark DIPT on source localization task against four methods: (1) **LPSI** (Wang et al., 2017): labels-based inference without explicit dynamics. (2) **OJC** (Zhu et al., 2017): SIR-based inference from partial observations. (3) **GCNSI** (Dong et al., 2019): GCN-based prediction of multiple sources. (4) **SLVAE** (Ling et al., 2022): variational modeling of source distributions.

Experimental configurations, implementation details, including tuned hyperparameters and training setup is provided in Appendix B.2. Our code is available at https://anonymous.4open.science/r/DIPT-A773

### 4.3 EXPERIMENTAL RESULTS

#### 4.3.1 PROPAGATION TREES PREDICTION PERFORMANCE

Table 1 summarizes the performance of DIPT against two baselines for propagation tree identification. DIPT consistently outperforms both across all eight datasets on path precision and Jaccard index. The performance gap is especially large on denser graphs like Power Grid, IDSS, Memetracker, where baseline methods struggle due to complex propagation dynamics and real-world noise. On sparser datasets (Cora-ML, CiteSeer), the margin is relatively smaller but still favorable. This demonstrates DIPT's robustness in both synthetic and real-world settings, enabled by its

Table 2: Source Localization performance comparison based on F1 score and AUC (best in bold).

| Method | Cora-ML | | Memetracker | | CiteSeer | | Power Grid | | IDSS | | Twitter | | Weibo | | Pol | |
|---|---|---|---|---|---|---|---|---|---|---|---|---|---|---|---|---|
| | F1 | AUC | F1 | AUC | F1 | AUC | F1 | AUC | F1 | AUC | F1 | AUC | F1 | AUC | F1 | AUC |
| LPSI | 0.301 | 0.592 | 0.014 | 0.529 | 0.306 | 0.598 | 0.474 | **0.934** | 0.037 | 0.540 | 0.201 | 0.578 | 0.227 | 0.552 | 0.193 | 0.575 |
| OJC | 0.121 | 0.534 | 0.026 | 0.517 | 0.117 | 0.530 | 0.153 | 0.501 | 0.028 | 0.520 | 0.084 | 0.491 | 0.077 | 0.470 | 0.082 | 0.490 |
| GCNSI | 0.401 | 0.687 | 0.035 | 0.422 | 0.387 | 0.680 | 0.330 | 0.639 | 0.045 | 0.430 | 0.287 | 0.603 | 0.265 | 0.567 | 0.261 | 0.600 |
| SLVAE | 0.764 | 0.831 | 0.488 | 0.624 | 0.749 | 0.825 | 0.797 | 0.879 | 0.494 | 0.630 | 0.253 | 0.578 | 0.209 | 0.547 | 0.209 | 0.576 |
| DDMIX | 0.221 | 0.247 | 0.022 | 0.417 | 0.215 | 0.250 | 0.280 | 0.340 | 0.029 | 0.425 | 0.176 | 0.464 | 0.155 | 0.439 | 0.168 | 0.470 |
| DITTO | 0.384 | 0.660 | 0.115 | 0.545 | 0.369 | 0.660 | 0.488 | 0.700 | 0.151 | 0.555 | 0.230 | 0.595 | 0.209 | 0.555 | 0.239 | 0.585 |
| DDMSL | 0.750 | **0.873** | **0.515** | **0.641** | 0.742 | 0.870 | **0.831** | 0.866 | **0.527** | **0.645** | 0.313 | 0.625 | 0.381 | 0.622 | 0.317 | 0.630 |
| DIPT | **0.839** | **0.881** | **0.518** | 0.629 | **0.832** | **0.880** | 0.828 | 0.864 | 0.525 | 0.630 | **0.421** | **0.648** | **0.439** | **0.658** | **0.429** | **0.655** |

Table 3: Performance of DIPT with varying proportions of partially observed propagation tree data during training.

| Data % | Cora-ML | | Memetracker | | CiteSeer | | Power Grid | | IDSS | | Twitter | | Weibo | | Pol | |
|---|---|---|---|---|---|---|---|---|---|---|---|---|---|---|---|---|
| | Prec. | Jac. | Prec. | Jac. | Prec. | Jac. | Prec. | Jac. | Prec. | Jac. | Prec. | Jac. | Prec. | Jac. | Prec. | Jac. |
| 10% | 0.671 | 0.504 | 0.633 | 0.463 | 0.662 | 0.495 | 0.683 | 0.518 | 0.437 | 0.279 | 0.595 | 0.430 | 0.567 | 0.410 | 0.582 | 0.422 |
| 20% | 0.707 | 0.546 | 0.651 | 0.482 | 0.671 | 0.504 | 0.718 | 0.560 | 0.451 | 0.291 | 0.612 | 0.445 | 0.579 | 0.425 | 0.599 | 0.439 |
| 30% | 0.720 | 0.562 | 0.662 | 0.494 | 0.695 | 0.532 | 0.759 | 0.611 | 0.453 | 0.292 | 0.628 | 0.461 | 0.597 | 0.442 | 0.618 | 0.455 |

edge-level influence modeling (Eq. 5). DIPT achieves its best results on Power Grid (68% path precision, 51.5% Jaccard), and handles the challenging IDSS dataset with 42.1% path precision—far surpassing DDMSL (12.1%) and DDMIX (10.9%). It also generalizes well to large-scale graphs, achieving strong performance on Twitter (58.1%, 41.8%) and Weibo (55.2%, 39.7%), where baselines considerably underperforms.

Figure 2 illustrates predicted trees on Memetracker[4] DDMSL and DDMIX exhibit more incorrect edge predictions (orange edges) in dense subgraphs. DIPT better handles such dense regions by learning influence between nodes, leading to more accurate edge selection. Additional visualizations are in Appendix F. On average, DIPT improves path precision and Jaccard index by 3.5× and 4.37×, respectively, over all three baselines.

### 4.3.2 SOURCE LOCALIZATION ACCURACY

We evaluate DIPT against six source localization methods, with results summarized in Table 2. DIPT performs competitively with DDMSL and SLVAE across all datasets and significantly outperforms LPSI, OJC, and GCNSI. DIPT achieves the highest F1 (0.839) and AUC (0.881) scores on Cora-ML and performs strongly on CiteSeer, consistent with its superior propagation tree accuracy on these less dense graphs. On IDSS, which exhibits varying seed distributions due to mobility patterns, DIPT matches DDMSL's performance (F1: 0.525, AUC: 0.630). It also outperforms all baselines on large-scale Twitter and Weibo graphs, achieving F1 scores of 0.421 and 0.439, respectively, demonstrating strong generalization to large networks. These results show that despite solving the harder task of full propagation tree inference, DIPT achieves source localization accuracy comparable to specialized methods. Precision and Recall metrics are included in Appendix E.2.

### 4.3.3 IMPACT OF PARTIAL OBSERVATIONS

As DIPT is primarily designed to learn propagation trees without edge supervision, Section 3.6 explores how incorporating partial tree information during training affects performance. Table 3 shows that providing 10%, 20%, and 30% of edge annotations leads to average path precision gains of 7.11%, 11.84%, and 15.16%, respectively. Similar improvements are observed on Twitter and Weibo; for example, Twitter precision rises from 0.581 (unsupervised) to 0.595 (10% supervision), highlighting DIPT's adaptability across graph scales. These gains stem from the supervised loss in Eq. 13, which enhances learning of influence patterns. However, since inference is based on maximizing infection likelihood, supervision impacts are non-linear with respect to annotation quantity.

---

[4]Every infected(pink) node has a parent. Some appear without incoming edges because their parent lies outside the displayed subgraph. Due to space constraints, nodes are subsampled for visualization

Table 4: Performance of DIPT under different ablation settings.

| Ablation | Cora-ML | | Memetracker | | CiteSeer | | Power Grid | | IDSS | | Twitter | | Weibo | | Pol | |
|---|---|---|---|---|---|---|---|---|---|---|---|---|---|---|---|---|
| | Prec. | Jac. | Prec. | Jac. | Prec. | Jac. | Prec. | Jac. | Prec. | Jac. | Prec. | Jac. | Prec. | Jac. | Prec. | Jac. |
| DIPT (a) | 0.388 | 0.235 | 0.407 | 0.255 | 0.422 | 0.267 | 0.437 | 0.276 | 0.329 | 0.206 | 0.368 | 0.248 | 0.362 | 0.243 | 0.372 | 0.252 |
| DIPT (b) | 0.519 | 0.350 | 0.489 | 0.326 | 0.511 | 0.343 | 0.607 | 0.435 | 0.371 | 0.227 | 0.495 | 0.335 | 0.489 | 0.330 | 0.502 | 0.341 |
| DIPT | **0.622** | **0.452** | **0.602** | **0.430** | **0.593** | **0.421** | **0.680** | **0.515** | **0.421** | **0.266** | **0.581** | **0.418** | **0.552** | **0.397** | **0.565** | **0.404** |

### 4.3.4 Diffusion State Reconstruction

Both DDMSL and DDMIX reconstruct node states at discrete time steps during the diffusion process, capturing an infection sequence that reflects the order in which nodes became infected (i.e., who was infected after whom). In contrast, DIPT not only recovers this infection order (induced from the inferred propagation tree) but also identifies the source of each infection (i.e., who infected whom), thereby reconstructing the propagation tree. To ensure a fair comparison, we evaluate all methods on node state reconstruction accuracy with respect to the infection order. As shown in Fig. 3, DIPT achieves an average mean squared error (MSE) that is 19.67% lower than DDMIX,

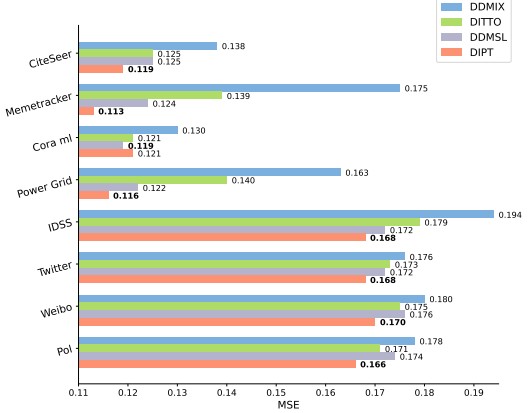

Figure 3: Comparison of DIPT, DITTO, DDMSL and DDMIX on reconstructed information diffusion.

5.2% lower than DITTO and 3.85% lower than DDMSL. These results highlight that, beyond recovering propagation tree, DIPT accurately models node-level infection dynamics as well.

### 4.3.5 Ablation Results

We conduct an ablation study to assess the contribution of each core component in DIPT. In the first variant, DIPT(a), the learned edge influence (Eq.5) is replaced with cosine similarity between node features. In DIPT(b), the model directly optimizes the joint objective in Eq.3 without alternating between tree inference and parameter updates during training; however, inference still uses the propagation tree inference described in Section 3.7. Results are reported in Table 4. Both ablations lead to notable performance drops. In DIPT(a), accuracy degrades significantly because edge-level influence is no longer learned—scores remain static, undermining the model's ability to capture propagation structure. On Cora-ML, DIPT(a) even underperforms DDMSL in path precision (Table 1), though it remains competitive on denser datasets where simple similarity still provides some signal. DIPT(b) performs better than DIPT(a) and continues to outperform all baselines, but its performance lags behind full DIPT. This highlights that while both components are important, learning local node influence is more critical than alternating optimization. **Additional Experimental Results** showing generalization, features sensitivity, and inference time analysis are documented in Appendix E.1 and C respectively.

## 5 Conclusion

Identification of propagation trees is a crucial yet underexplored task with significant applications in fields such as epidemiology and misinformation diffusion. In this paper, we introduce DIPT, a probabilistic framework designed to identify propagation trees from observed diffusion data. DIPT recursively models the diffusion process by learning influence probabilities across edges. The framework employs an alternating optimization approach to jointly learn both the propagation tree and the diffusion mechanism, without relying on direct observation of propagation paths during training. Extensive experiments on eight datasets demonstrate that DIPT consistently outperforms existing methods, achieving an average path precision of 58.2% and effectively identifying both propagation trees and diffusion sources.

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

## A  INFECTIOUS DISEASE SPREAD SIMULATION

We simulate the spread of an infectious disease across the 3,143 counties of the United States using a spatial compartmental Susceptible-Infectious-Recovered (SIR) model (Kermack & McKendrick, 1932). The simulation incorporates real-world human mobility data from SafeGraph (Kang et al., 2020; SafeGraph, Inc.), enabling region-level interactions based on observed population flows.

### SIMULATION SETUP

Each county $C$ is initialized with a susceptible population $S_C$ based on U.S. Census data (United States Census Bureau), with $I_C = R_C = 0$. Infected individuals follow a time-dependent reproduction number vector $P = [0.2, 0.3, 0.3, 0.2, 0.1, 0.1]$, where $P_i$ denotes the probability of infecting others on day $i$ of infection. The total reproduction number is $R_0 = \sum_i P_i = 1.2$, reflecting a pathogen with pandemic potential (Delamater et al., 2019).

When an individual in County $A$ becomes infectious, a newly infected individual is assigned to a County $B$ using a two-step mobility-informed sampling:

1. Sample County $X$ from County $A$'s outflow distribution.
2. Sample County $B$ from County $X$'s inflow distribution.

This models the scenario where an infected individual travels and interacts with someone who visited the same area.

### INFECTION SEEDING AND PROGRESSION

To initiate the simulation, we simulate infected air travelers arriving in high-traffic counties. From the 72 U.S. counties with major airports, two are randomly selected in each run, and 10 infected individuals are introduced. Their destinations are sampled based on the mobility outflows from

these counties, simulating infected passengers returning home. The simulation runs for 90 days. Each infected individual remains infectious for 6 days, transitioning from $I$ to $R$ afterward. Each transmission event reduces $S$ and increases $I$ in target county; recovery reduces $I$ and increases $R$.

OUTPUT FORMAT

Each simulation generates a **propagation forest**:

- **Nodes:** Infected individuals.
- **Edges:** "Who-infected-whom" links.

Since multiple initial seeds exist, the resulting structure is a forest, where each tree traces back to a unique root source. Across runs, simulations yield approximately 500–1,000 infected counties and 12k–48k infected individuals. The dataset can be accessed at following link: https://anonymous.4open.science/r/Compartmental-Infectious-Disease-Simulation-4F7C

Table 5: Hyperparameter settings of different algorithms.

| Algorithms | Hyper-parameter | Cora ML | Memetracker | Citeseer | Power Grid | IDSS | Search space / Description |
|---|---|---|---|---|---|---|---|
| DIPT | Propagation steps ($K$) | 25 | 30 | 25 | 10 | 20 | $[min = 10, max = 50, step = 5]$ |
| | Training epochs ($E$) | 100 | 100 | 100 | 100 | 100 | Total training iterations (Fixed) |
| | Latent dimension ($z$) | 300 | 300 | 300 | 300 | 300 | Size of latent space for seed prior (Fixed) |
| | Learning rate | $[4 \times 10^{-4}]$ | $[4 \times 10^{-4}]$ | $[4 \times 10^{-4}]$ | $[4 \times 10^{-4}]$ | $[4 \times 10^{-4}]$ | $[4 \times 10^{-4}, 6 \times 10^{-4}, 1 \times 10^{-3}]$ |
| | Regularization weight ($\lambda$) | 0.5 | 0.5 | 0.5 | 0.5 | 0.5 | $[min = 0.3, max = 0.7, step = 0.1]$ $\mathcal{L}_{\text{diff}}$ |
| | Regularization weight ($\mu$) | 0.2 | 0.3 | 0.2 | 0.3 | 0.6 | $[min = 0.1, max = 0.7, step = 0.1]$ |
| | Inference regularization ($\gamma$) | 0.1 | 0.1 | 0.1 | 0.1 | 0.4 | $[min = 0.05, max = 0.4, step = 0.05]$ |
| DITTO | Training epochs ($E$) | 100 | 100 | 100 | 100 | 100 | Same as other baselines |
| | $\beta$ optimization iterations ($I$) | 500 | 500 | 500 | 500 | 500 | Fixed (proposal optimization) |
| | Proposal network $Q_\theta$ | 3-layer GNN + 2-layer MLP (hidden size 16) | | | | | Fixed architecture |
| | Training iterations ($J$) | 300 | 300 | 300 | 300 | 300 | Total proposal training iterations[min-200, max=600] |
| DDMSL | Initial learning rate | $2 \times 10^{-3}$ | $2 \times 10^{-3}$ | $2 \times 10^{-3}$ | $2 \times 10^{-3}$ | $3 \times 10^{-3}$ | $[2 \times 10^{-3}, 4 \times 10^{-3}, 5 \times 10^{-3}]$ |
| | Learning rate decline interval | [1200,1500] | [200,1000] | [500,1200] | [500,1200] | [200,500,800,1200] | *LOSS* curve-based schedule |
| | $\alpha$ in SIR model | 0.4 | 0.6 | 0.4 | 0.4 | 0.4 | $[min = 0.3, max = 0.7, step = 0.05]$ |
| | $\alpha$ in SI model | 0.4 | 0.45 | 0.4 | 0.4 | 0.4 | $[min = 0.3, max = 0.7, step = 0.05]$ |
| DDMIX | $\alpha$ | 0.5 | 0.5 | 0.5 | 0.5 | 0.5 | $[min = 3, max = 9, step = 1]$ |
| | Learning rate | $2 \times 10^{-3}$ | $2 \times 10^{-3}$ | $2 \times 10^{-3}$ | $2 \times 10^{-3}$ | $3 \times 10^{-3}$ | $[2 \times 10^{-3}, 4 \times 10^{-3}, 5 \times 10^{-3}]$ |
| | Epoch | 100 | 100 | 100 | 100 | 100 | Fixed |

# B IMPLEMENTATION DETAILS

We begin by projecting node features into a lower-dimensional space using a three-layer MLP. To enrich these representations with structural context, we apply a three-layer GNN that integrates graph topology into the node embeddings. A cross-attention module then fuses the transformed features, and the resulting representations are passed through a two-layer MLP to compute pairwise influence scores, following the scaled dot-product attention formulation. Model training is performed with the Adam optimizer at a learning rate of $\eta$ for $T$ epochs. To learn a prior over seed nodes, we employ a three-layer MLP with nonlinear transformations in both the encoder $q_{\phi_E}(z|x)$ and decoder $p_{\phi_D}(s|z)$. This component is trained with a fixed learning rate of 0.005 over 500 epochs for all datasets. During inference, we run 100 iterations consistently across datasets. The implementation code is provided as part of the supplementary material. For starting 5 iterations we use beam search i.e. instead of one most probable path to each node chose top three for stable learning.

## B.1 PRACTICAL INFERENCE OBJECTIVE DERIVATION

Equation (16) defines the ideal inference objective as a joint maximization over latent sources $s$ and the propagation structure $\mathcal{T}$. However, this objective requires marginalizing over all source configurations and latent codes:

$$\mathcal{L}_{\text{infer}} = \mathbb{E}_{s \sim p_\phi(s|z)} \left[ \log P_\psi(\mathbf{y} \mid s, \mathcal{T}, G) \right]$$

This marginalization is computationally infeasible due to the exponential size of the source space and propagation paths. Therefore, we simplify the inference using two likelihoods:

- A **Gaussian likelihood** for the continuous infection vector $\mathbf{y} \in [0,1]^{|V|}$, enabling mean squared error as a surrogate.

- A **Bernoulli likelihood** for the binary seed vector $s \in \{0,1\}^{|V|}$, parameterized by a decoder $f_{\phi_D}(z) \in [0,1]^{|V|}$.

---

**Algorithm 2** DIPT *Inference*: Latent Optimization & Tree Reconstruction (compact)

---

**Input:** $G = (V, E)$, features $F$, observed infections $\mathbf{y}$, learned $(\phi, \psi)$, steps $K$, iters $E$, lr $\eta$, reg $\gamma$
**Output:** $P_\phi(s \mid \hat{z})$, propagation tree $\mathcal{T}^*$
Initialize $\hat{z} \leftarrow \bar{z}$; **for** $e = 1$ **to** $E$ **do**

    // Seed prior from latent
    $P_\phi(s \mid \hat{z})$; sample/decoding $s \sim P_{\phi_D}(s \mid \hat{z})$ ;          // Sec. 3.5, equation 15
    // Edge transmissions
    $I_{uv} = f_\psi(F_u, F_v), \; \forall (u, v) \in E$ ;               // equation 5
    // Monotone propagation
    $P_{\text{inf}}^{(0)}(v) = P_\phi(s_v \mid \hat{z})$; **for** $k = 1$ **to** $K$ **do**
        $P_{\text{inf}}^{(k)}(v) = \max\big(P_{\text{inf}}^{(k-1)}(v), \; \max_{u \in \mathcal{C}(v)} P_{\text{inf}}^{(k-1)}(u) \, I_{uv}\big), \forall v$
    **end**
    // Tree extraction (acyclic by construction)
    $p_v = \arg\max_{u \in \mathcal{C}(v)} P_{\text{inf}}^{(K)}(u) \, I_{uv}$ for $y_v = 1, v \notin s$; $\mathcal{T}^* = \{(p_v, v)\}$ ;    // cf. Eq. (10)
    // Likelihood and latent update
    $P_\psi(\mathbf{y} \mid s, \mathcal{T}^*, G) = \prod_{v \notin s} P_\psi(y_v \mid y_{p_v})$ ;         // equation 4
    $\mathcal{L}_{\text{inf}} = -\log P_\psi(\mathbf{y} \mid s, \mathcal{T}^*, G) + \gamma \|\hat{z} - \bar{z}\|^2$ ;      // equation 15
    $\hat{z} \leftarrow \hat{z} - \eta \nabla_{\hat{z}} \mathcal{L}_{\text{inf}}$
**end**
**return** $P_\phi(s \mid \hat{z})$, $\mathcal{T}^*$

---

- The Bernoulli term involves products over all nodes:

$$\prod_{i=1}^{|V|} f_{\phi_D}(z_i)^{s_i} (1 - f_{\phi_D}(z_i))^{1-s_i}$$

which becomes numerically unstable when $|V|$ is large. To address this, we apply the log-sum-exp trick to derive a numerically stable surrogate objective:

$$\log \left[ \sum_j \exp \left( \sum_i \log f_{\phi_D}(z_i^j)^{s_i} (1 - f_{\phi_D}(z_i^j))^{1-s_i} \right) \right]$$

This log-sum-exp approximation yields a tractable and stable objective.

Hence, we adopt the following practical objective in Eq. (17), combining the Gaussian reconstruction error over $\mathbf{y}$, the Bernoulli loss over $s$, and a regularization penalty over the latent code $z$:

$$\min_{\hat{z}} \quad \|\hat{\mathbf{y}} - \mathbf{y}\|^2 - \sum_i \log f_{\phi_D}(\hat{z}_i)^{s_i} (1 - f_{\phi_D}(\hat{z}_i))^{1-s_i} + \lambda \|\hat{z} - \bar{z}\|^2$$

## B.2 HYPERPARAMETER DETAILS

Tuned hyperparameters for baselines and DIPT are summarized in Table 5. For baselines, limited search space exploration is done as in original work.

## B.3 FURTHER DATASET DETAILS

The further description of the datasets used for the experiments are shown as below:

- **Cora-ML** Rossi & Ahmed (2015). This network contains computer science research papers, where each node represents a paper and each edge indicates that one paper cites another.

- **Power Grid** Watts & Strogatz (1998). This is the topology network of the Western States Power Grid of the US. An edge represents a power supply line, and a node is either a generator, a transformer, or a substation.

- **Memetracker** Leskovec et al. (2009). MemeTracker tracks the posts that appear most frequently over time across the entire online news spectrum. The propagation of each story is represented as one diffusion cascade.

- **CiteSeer** Caragea et al. (2014). This is a citation network of research papers, where each node represents a paper, and edges indicate citation relationships. Papers are classified into different categories based on their research topics.

- **IDSS**. This is a mobility network of US counties based on real mobility data and is explained in detail in Appendix A.

- **Twitter** Weng et al. (2013). This dataset contains public English-language tweets published between Mar 24 and Apr 25, 2012. Nodes are users; the global user–user graph aggregates three relations: reciprocal follow ties, retweets, and mentions. We treat each hashtag and its adopters as an independent cascade, with cascade edges induced by the three relations above.

- **Weibo** Cao et al. (2017). Weibo is a large microblogging platform in China. Nodes are users, and the global graph is constructed from user retweeting relationships. Each original post and its retweets form a retweet cascade.

- **Pol** Conover et al. (2011) is a temporal retweet network about a U.S. political event. It is an SI-like diffusion, because when a user retweets or is retweeted, they must have known about the event.

## C    TIME COMPLEXITY ANALYSIS

**Time & space complexity.**    DIPT has two components: propagation and model training. Let $K$ be the propagation steps (upper-bounded by diffusion depth / graph diameter), and let $C_{\text{model}} = \sum_\ell d_{\ell-1} d_\ell$ be the per-edge MLP cost (e.g., for a 2-layer MLP $2d \to h \to 1$, $C_{\text{model}} = \mathcal{O}(2dh + h)$).

**Propagation.** Each step is a sparse mat-vec plus a max-reduction over neighbors: $\mathcal{O}(E)$ per step $\Rightarrow \mathcal{O}(KE)$ per pass, with masking/parent-pick in $\mathcal{O}(E)$. Worst case (path graph) gives $\mathcal{O}(VE)$ when $K = \Theta(V)$; in practice we cap $K$, so it is $\mathcal{O}(KE)$.

**Influence model training.** Per epoch: edge scoring $\mathcal{O}(E\,C_{\text{model}})$ (fwd+bwd) + one propagation/loss eval $\mathcal{O}(KE)$, so $\mathcal{O}\big(E(C_{\text{model}} + K)\big)$ per epoch and $\mathcal{O}\big(\mathcal{E}\,E(C_{\text{model}} + K)\big)$ over $\mathcal{E}$ epochs. On GPU this parallelizes over edges, giving near-linear wall-clock in $E$ for fixed $C_{\text{model}}, K$.

**Space.** Graph and influence scores $\mathcal{O}(E)$; node vectors $\mathcal{O}(V)$; parameters $\mathcal{O}(C_{\text{model}})$.

**Comparison with DDMSL/DDMIX.** DDMSL scales with diffusion steps and GNN depth, e.g., $\mathcal{O}\big(\mathcal{E} \cdot V \cdot T \cdot F^2 + \mathcal{E} \cdot L \cdot E \cdot F\big)$ for $T$ time steps, $L$ GCN layers, feature width $F$. DDMIX also grows with $T$ (despite a lightweight VAE). In contrast, DIPT avoids observed timestamps (it only uses $K$ propagation passes) and uses edge-local updates, making it significantly more efficient on large, sparse graphs.

Table 6: Inference Time Comparison (in seconds) across methods and datasets for propagation tree inference task.

| Method | Cora-ML | Power Grid | Memetracker |
|--------|---------|------------|-------------|
| DIPT   | 11.1    | 17.2       | 47.8        |
| DDMSL  | 16.1    | 21.7       | 49.3        |
| DITTO  | 54.5    | 67.4       | 100.9       |
| DDMIX  | 9.1     | 15.3       | 44.2        |

## D    THEORETICAL PROOFS AND ANALYSIS

### D.1    MONOTONICITY

**Theorem 1** (Infection Monotonicity of DIPT). *Let $S \subseteq T$ be two seed sets with one-hot vectors $x_S, x_T \in \{0,1\}^{|V|}$. Let $P_{inf,S}^{(k)}$ and $P_{inf,T}^{(k)}$ be DIPT infection probability vectors at step $k$ when initialized with $S$ and $T$, respectively, under: (i) $I$ is elementwise nonnegative; (ii) $P_{inf,S}^{(0)} \preceq P_{inf,T}^{(0)}$ (e.g., $P_{inf}^{(0)}(v) = \alpha\, x(v)$ for a fixed $\alpha > 0$, or any initialization that increases elementwise with the seed set).*

| Training Data | Method | Cora-ML | | Power Grid | | IDSS | |
|---|---|---|---|---|---|---|---|
| | | Jaccard Index | Path Precision | Jaccard Index | Path Precision | Jaccard Index | Path Precision |
| Original | DIPT | 0.452 | 0.622 | 0.515 | 0.680 | 0.266 | 0.421 |
| | DDMSL | 0.259 | 0.412 | 0.069 | 0.130 | 0.064 | 0.121 |
| | DDMIX | 0.195 | 0.327 | 0.031 | 0.081 | 0.057 | 0.109 |
| Small World | DIPT | 0.461 | 0.632 | 0.509 | 0.663 | 0.268 | 0.424 |
| | DDMSL | 0.255 | 0.381 | 0.060 | 0.118 | 0.034 | 0.083 |
| | DDMIX | 0.190 | 0.313 | 0.020 | 0.068 | 0.019 | 0.026 |
| ER | DIPT | 0.256 | 0.317 | 0.433 | 0.634 | 0.136 | 0.229 |
| | DDMSL | 0.101 | 0.165 | 0.039 | 0.090 | 0.081 | 0.128 |
| | DDMIX | 0.091 | 0.142 | 0.033 | 0.088 | 0.056 | 0.107 |
| BA Tree | DIPT | 0.333 | 0.472 | 0.509 | 0.642 | 0.211 | 0.407 |
| | DDMSL | 0.199 | 0.381 | 0.044 | 0.118 | 0.066 | 0.128 |
| | DDMIX | 0.182 | 0.353 | 0.033 | 0.084 | 0.053 | 0.101 |

Table 7: Generalization performance comparison of DIPT, DDMSL, and DDMIX across three datasets under various synthetic graph structure conditions.

*With the update*

$$P_{inf}^{(k)} = \max\left( P_{inf}^{(k-1)}, P_{inf}^{(k-1)} I \right) \quad (k \geq 1),$$

*applied elementwise, it holds that*

$$P_{inf,S}^{(k)} \preceq P_{inf,T}^{(k)} \quad \text{for all } k \geq 0.$$

*Proof.* Base case: by assumption, $P_{inf,S}^{(0)} \preceq P_{inf,T}^{(0)}$. Inductive step: assume $P_{inf,S}^{(k-1)} \preceq P_{inf,T}^{(k-1)}$. Since $I \geq 0$, we have $P_{inf,S}^{(k-1)} I \preceq P_{inf,T}^{(k-1)} I$. The map $F(P) = \max(P, PI)$ is order-preserving (monotone) because both the linear map $P \mapsto PI$ and the elementwise max are monotone. Hence

$$P_{inf,S}^{(k)} = F\big(P_{inf,S}^{(k-1)}\big) \preceq F\big(P_{inf,T}^{(k-1)}\big) = P_{inf,T}^{(k)}.$$

$\square$

**Assumption Justification in DIPT.** (1) The edge influence scores $I_{uv} = f_\psi(F_u, F_v)$ are passed through a sigmoid/softmax, ensuring $I_{uv} \in [0, 1]$, hence $I \geq 0$. (2) Initialization uses the learned source prior $P_\phi(s)$; if $S \subseteq T$, then the one-hot vectors satisfy $x_S \preceq x_T$, and consequently $P_{inf,S}^{(0)} \preceq P_{inf,T}^{(0)}$.

### D.2 CONVERGENCE OF ALTERNATING OPTIMIZATION IN DIPT

DIPT's generative model samples seed nodes $s \sim P_\phi(s)$ and constructs a propagation tree $\mathcal{T} = T(s; \psi)$, as described in Section 3.4. Since training operates on inferred propagation trees $\mathcal{T}$ rather than directly on $s$, we define the induced joint distribution over observed infections $\mathbf{y}$ and propagation trees $\mathcal{T}$ as:

$$p(\mathbf{y}, \mathcal{T}; \theta) := p_\phi(\mathcal{T}) \cdot p_\psi(\mathbf{y} \mid \mathcal{T}),$$
$$\text{where } p_\phi(\mathcal{T}) = \sum_{s:T(s;\psi)=\mathcal{T}} P_\phi(s).$$

Let $\theta = (\phi, \psi)$ denote model parameters, and let $\mathcal{T} \in \mathcal{T}_G$ be a valid propagation tree. The marginal log-likelihood is:

$$\mathcal{L}(\theta) := \log p(\mathbf{y}; \theta) = \log \sum_{\mathcal{T} \in \mathcal{T}_G} p(\mathbf{y}, \mathcal{T}; \theta).$$

Since the number of trees $\mathcal{T}_G$ grows exponentially, this marginal is intractable. DIPT instead uses the following alternating optimization:

- **Tree update:** $\mathcal{T}^{(t)} := \arg\max_{\mathcal{T}} \log p(\mathbf{y}, \mathcal{T}; \theta^{(t)})$

- **Parameter update:** $\theta^{(t+1)} := \arg\max_{\theta} \log p(\mathbf{y}, \mathcal{T}^{(t)}; \theta)$

We now show that this procedure converges to a coordinate-wise stationary point of the joint objective.

**Theorem 2.** *Let $\{\theta^{(t)}, \mathcal{T}^{(t)}\}_{t=1}^{\infty}$ be the sequence generated by DIPT. Then:*

$$\mathcal{L}(\theta^{(t+1)}) \geq \mathcal{L}(\theta^{(t)}),$$

*and the sequence $\{\mathcal{L}(\theta^{(t)})\}$ converges. Moreover, the sequence $(\theta^{(t)}, \mathcal{T}^{(t)})$ converges to a coordinate-wise stationary point of the joint log-probability $\log p(\mathbf{y}, \mathcal{T}; \theta)$.*

*Proof.* Define a point-mass distribution over trees: $Q^{(t)}(\mathcal{T}) := \delta(\mathcal{T} = \mathcal{T}^{(t)})$. By Jensen's inequality, for any distribution $Q$ over $\mathcal{T}_G$:

$$\log p(\mathbf{y}; \theta) \geq \mathbb{E}_{Q(\mathcal{T})}[\log p(\mathbf{y}, \mathcal{T}; \theta)] + \mathcal{H}(Q) =: \mathcal{J}(Q, \theta),$$

where $\mathcal{H}(Q)$ denotes the entropy of $Q$. For a deterministic $Q^{(t)}$, we get:

$$\mathcal{J}(Q^{(t)}, \theta) = \log p(\mathbf{y}, \mathcal{T}^{(t)}; \theta).$$

From the tree update step:

$$\mathcal{T}^{(t)} = \arg\max_{\mathcal{T}} \log p(\mathbf{y}, \mathcal{T}; \theta^{(t)}) \Rightarrow \mathcal{L}(\theta^{(t)}) \geq \mathcal{J}(Q^{(t)}, \theta^{(t)}).$$

From the parameter update:

$$\theta^{(t+1)} = \arg\max_{\theta} \mathcal{J}(Q^{(t)}, \theta) \Rightarrow \mathcal{J}(Q^{(t)}, \theta^{(t+1)}) \geq \mathcal{J}(Q^{(t)}, \theta^{(t)}).$$

Combining the above:

$$\mathcal{L}(\theta^{(t+1)}) \geq \mathcal{J}(Q^{(t)}, \theta^{(t+1)}) \geq \mathcal{J}(Q^{(t)}, \theta^{(t)}) \leq \mathcal{L}(\theta^{(t)}).$$

Hence, $\mathcal{L}(\theta^{(t)})$ is non-decreasing. If we assume that $\mathcal{L}(\theta)$ is continuous and upper bounded, which holds in our case since (i) the parameter space of DIPT is compact due to bounded optimization ranges (e.g., for $\lambda$, $\mu$, learning rate), and (ii) the log-likelihood is computed over a finite graph with bounded probabilities, ensuring $\log p(\mathbf{y}; \theta)$ is finite.

Since DIPT alternates exact maximization over discrete (tree) and continuous (parameter) variables, and each step performs coordinate-wise maximization, the limit point is a coordinate-wise stationary point of $\log p(\mathbf{y}, \mathcal{T}; \theta)$. □

*Note:* DIPT does not perform full marginalization over latent trees; it instead maximizes a lower bound via greedy MAP inference. This proof establishes convergence as a special case of block coordinate ascent applied to a non-convex function, which is widely used in variational and EM-like training settings.

Table 8: Source Localization performance (Recall and Precision) — Supplementary metrics for Table 2, including Twitter, Weibo, and Pol.

| Method | Cora-ML | | Memetracker | | CiteSeer | | Power Grid | | IDSS | | Twitter | | Weibo | | Pol | |
|---|---|---|---|---|---|---|---|---|---|---|---|---|---|---|---|---|
| | RE | PR | RE | PR | RE | PR | RE | PR | RE | PR | RE | PR | RE | PR | RE | PR |
| LPSI | 0.217 | 0.492 | 0.292 | 0.007 | 0.225 | 0.480 | 0.495 | 0.455 | 0.280 | 0.020 | 0.199 | 0.175 | 0.202 | 0.188 | 0.205 | 0.182 |
| OJC | 0.119 | 0.123 | 0.022 | 0.031 | 0.115 | 0.118 | 0.287 | 0.104 | 0.025 | 0.033 | 0.097 | 0.072 | 0.083 | 0.069 | 0.090 | 0.075 |
| GCNSI | 0.456 | 0.357 | 0.234 | 0.019 | 0.440 | 0.345 | 0.335 | 0.325 | 0.245 | 0.025 | 0.298 | 0.237 | 0.271 | 0.222 | 0.285 | 0.240 |
| SLVAE | 0.719 | 0.814 | 0.518 | **0.461** | 0.700 | 0.805 | 0.780 | 0.815 | 0.520 | **0.470** | 0.256 | 0.174 | 0.212 | 0.151 | 0.240 | 0.185 |
| DDMIX | 0.210 | 0.232 | 0.023 | 0.021 | 0.205 | 0.225 | 0.345 | 0.235 | 0.030 | 0.028 | 0.182 | 0.145 | 0.160 | 0.131 | 0.190 | 0.150 |
| DITTO | 0.365 | 0.405 | 0.145 | 0.095 | 0.355 | 0.385 | 0.525 | 0.455 | 0.220 | 0.115 | 0.248 | 0.215 | 0.225 | 0.195 | 0.255 | 0.225 |
| DDMSL | 0.758 | 0.742 | **0.618** | 0.441 | 0.750 | 0.735 | 0.763 | **0.913** | **0.625** | 0.455 | 0.322 | 0.309 | 0.374 | 0.330 | 0.335 | 0.300 |
| DIPT | **0.856** | **0.823** | 0.607 | 0.452 | **0.850** | **0.815** | **0.781** | 0.882 | 0.610 | 0.460 | **0.438** | **0.407** | **0.464** | **0.419** | **0.450** | **0.410** |

### D.3 THEORETICAL ANALYSIS OF ERROR CORRECTION IN PROPAGATION TREE

DIPT alternates between updating model parameters and re-inferring the propagation tree. While path reconstruction is more error-prone than source localization (due to compounding parent choices), the alternating scheme enables recovery from earlier mistakes.

Let $\Phi^{(k)} = (\phi^{(k)}, \psi^{(k)})$ denote parameters at iteration $k$, and $\mathcal{T}^{(k)}$ the inferred tree. Updates proceed as

$$\Phi^{(k+1)} = \arg\max_{\Phi} \mathcal{L}(\Phi, \mathcal{T}^{(k)}), \qquad \mathcal{T}^{(k+1)} = \arg\max_{\mathcal{T}} \mathcal{L}(\Phi^{(k+1)}, \mathcal{T}).$$

Suppose in $\mathcal{T}^{(k)}$ a node $v$ is incorrectly assigned parent $p(v)$. If the updated parameters $\Phi^{(k+1)}$ increase the margin in favor of a better parent $p'(v)$, i.e.

$$\Delta = \log\left(P_{\mathrm{inf}}^{(K)}(p'(v)) \, I_{p'(v),v}\right) - \log\left(P_{\mathrm{inf}}^{(K)}(p(v)) \, I_{p(v),v}\right) > 0,$$

then the next tree update will assign $p'(v)$ as parent of $v$. Hence DIPT can correct earlier structural errors.

This correction is a direct consequence of likelihood-based parent selection (Eq. 10) combined with the monotone training objective: once a higher-likelihood edge is preferred, it is retained in subsequent iterations. This explains why path reconstruction improves steadily over epochs, despite noisy initial assignments.

### D.4 ACYCLICITY

**Theorem 3** (Acyclicity under strict update). *Let $P^{(k)}(\cdot)$ be the infection probabilities at step $k$ and let a non-source node $v$ adopt a parent $u$ only when the* strict gain *condition holds at the moment of adoption:*

$$P_{\psi}(v \mid u) \, P^{(\mathrm{pre})}(u) \; > \; P^{(\mathrm{pre})}(v), \tag{16}$$

*where $P^{(\mathrm{pre})}$ denotes values immediately* before *updating $v$. Then the directed parent map $\{(p_v, v)\}$ is acyclic.*

*Proof.* Assume, for contradiction, that the final parent map contains a directed cycle $u_1 \to u_2 \to \cdots \to u_m \to u_1$. For each edge $(u_i \to u_{i+1})$ on the cycle, let the adoption of $u_i$ as the parent of $u_{i+1}$ occur at some step, and evaluate equation 16 at that instant:

$$P_{\psi}(u_{i+1} \mid u_i) \, P^{(\mathrm{pre})}(u_i) \; > \; P^{(\mathrm{pre})}(u_{i+1}) \qquad (i = 1, \ldots, m).$$

Multiplying these $m$ strict inequalities gives

$$\prod_{i=1}^{m} P_{\psi}(u_{i+1} \mid u_i) \; \cdot \; \prod_{i=1}^{m} P^{(\mathrm{pre})}(u_i) \; > \; \prod_{i=1}^{m} P^{(\mathrm{pre})}(u_{i+1}).$$

The two products over $P^{(\mathrm{pre})}(\cdot)$ are identical up to index renaming (they contain the same set $\{u_1, \ldots, u_m\}$), hence they cancel, yielding

$$\prod_{i=1}^{m} P_{\psi}(u_{i+1} \mid u_i) \; > \; 1.$$

But every influence probability satisfies $0 < P_{\psi}(\cdot \mid \cdot) \le 1$, so the product on the left cannot exceed 1—a contradiction. Therefore no directed cycle exists. $\square$

**Remarks.** (i) The proof requires the *strict* gain rule equation 16 at the instant of adoption; this is exactly what our update uses (nodes only update when the candidate margin is strictly larger than the current value).

Table 9: DIPT robustness to feature perturbations (CORA-ML).

| Setting | Path Precision |
|---|---|
| Baseline (True Features) | 0.622 |
| Gaussian Noise (SNR = 20 dB) | 0.528 |
| Gaussian Noise (SNR = 10 dB) | 0.433 |
| Feature Dropout (50% dropped) | 0.405 |

## E ADDITIONAL EXPERIMENT

### E.1 GENERALIZATION ACROSS DYNAMIC GRAPHS

We evaluate the generalization performance of DIPT across three datasets by training all models on original dataset and testing on the synthetic topologies without retraining. This setting assesses the model's ability to transfer learned diffusion dynamics across networks with different topologies and scales. Results in Table 7 show that DIPT generalizes well across datasets.

### E.2 SOURCE LOCALIZATION RESULTS ON PRECISION AND RECALL EVALUATION METRICS

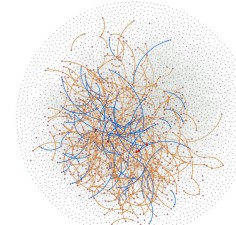 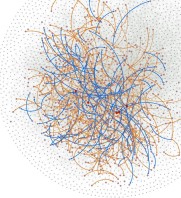 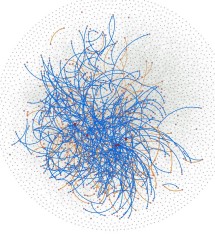 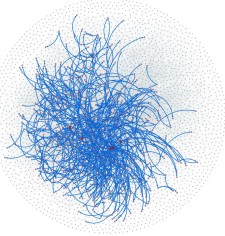

a) DDMIX        b) DDMSL        c) DIPT        d) Ground Truth

Figure 4: Comparison of predicted propagation tree edges with ground truth for the IDSS dataset. Source nodes are in red, infected nodes in pink.

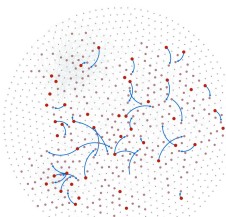 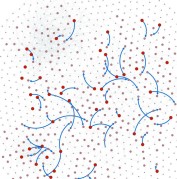 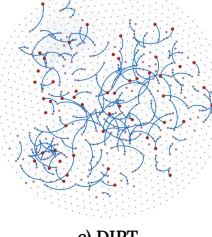 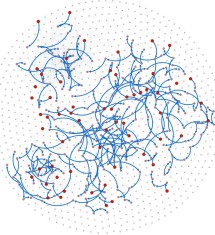

a) DDMIX        b) DDMSL        c) DIPT        d) Ground Truth

Figure 5: Comparison of correctly predicted propagation tree edges (blue) with ground truth for the Cora-ML dataset. Source nodes are in red, infected nodes in pink. Only correctly predicted edges are shown for clarity, with the total number of predicted edges being the same across all methods.

The source localization results on Precision and Recall metrics across all five datasets is shown in Table 8.

### E.3 ROBUSTNESS TO NOISY OR MISSING NODE FEATURES

We assess the robustness of DIPT to perturbations in node features, evaluating its capacity to infer propagation structure under noisy or incomplete information. In particular, we replace clean node features with (a) additive Gaussian noise at different signal-to-noise ratios (SNR = 20 dB and 10

dB), and (b) random feature dropout (50% of features masked). Results on the CORA-ML dataset (Table 9) show that while performance degrades under heavier perturbation, DIPT retains meaningful path-level inference. Notably, even with SNR as low as 10 dB or 50% dropout, the model still produces non-trivial propagation structure, highlighting its resilience to feature corruption.

### E.4    ROBUSTNESS TO DENSE SOURCE DISTRIBUTIONS

We also evaluate DIPT under settings with non-sparse source distributions. While real-world diffusion often begins from a small number of seeds, certain domains (e.g., mobility-driven or adversarial scenarios) may exhibit high source density. Since DIPT models a forest of trees rooted at seed nodes, its inference complexity scales linearly with the number of sources. To quantify its behavior in dense settings, we vary the percentage of source nodes in each cascade within the IDSS dataset. As expected, both path precision and Jaccard similarity decline gradually with increasing source count, reflecting the increased structural ambiguity in denser propagation forests. However, DIPT remains competitive, demonstrating its flexibility in adapting to varying sparsity regimes.

Table 10: Performance on IDSS with varying number of source nodes.

| Source % | Path Precision | Jaccard Similarity |
|:---:|:---:|:---:|
| 5 | 0.675 | 0.422 |
| 10 | 0.642 | 0.395 |
| 15 | 0.605 | 0.361 |
| 20 | 0.583 | 0.338 |

## F    ADDITIONAL VISUALIZATIONS

We provide additional visualizations of propagation tree identification for the Cora-ML and simulated IDSS dataset(Fig. 4 & 6). Blue edges represent correctly identified propagation edges, while orange edges indicate incorrectly predicted ones. Due to panel space limited width, two baselines and DIPT are visualized against Ground Truth.

## G    RELATED METHODS ON ALTERNATING OPTIMIZATION

**Alternating Optimization in Graph Problems** is used for tasks involving latent variables. Xiao et al. (2021) apply it in GNNs to learn personalized propagation strategies, while GLEM (Zhao et al., 2022) alternates between language models and GNNs for scalable learning on text-attributed graphs. MLCO (Wang et al., 2021) uses bi-level optimization for joint structure and reasoning. These works demonstrate the effectiveness of alternating updates in jointly refining model parameters and latent structures. Inspired by this, our method uses an alternating optimization to jointly infer propagation trees and learn diffusion dynamics.

