# OpenReview forum: "DIPT: Deep Identification of Propagation Trees in Graph Diffusion"
_ICLR.cc/2026/Conference — ICLR 2026 Conference Withdrawn Submission_

### Official Review · Reviewer_HTNY · 2025-10-23

**Soundness:** 2
**Presentation:** 2
**Contribution:** 2
**Rating:** 4
**Confidence:** 2

**Summary:**

This paper proposes a probabilistic framework Deep Identification of Propagation Trees,  to infer propagation trees from final observed node diffusion states, without knowledge of the diffusion mechanism. Empirical results across eight real-world datasets demonstrate that it  outperforms existing approaches in reconstructing propagation trees.

**Strengths:**

This paper recover actual ”who-infected-whom” propagation tree for a specific diffusion instance, which is limited in prior woks.

**Weaknesses:**

It is not clear why the path indentification works can't resolve this problem.

**Questions:**

How to sample the node set s from P(s)? What is the complexity?

---

> ### Author Response · Authors · 2025-11-28
>
> We’d like to thank the reviewer for your time and effort in reviewing our paper. We address your raised points as follows.
>
> > W1: It is not clear why the path indentification works can't resolve this problem.
>
> Prior path identification methods operate in a different setting and fall into two broad categories.
>
> 1) Classical approaches estimate global diffusion graphs or global edge parameters from many cascades, relying on temporal information or fixed diffusion models. They recover a global influence structure, not the instance level parent assignment for a specific infected state. These methods are therefore not designed to explain the who-infected-whom relations for a single final snapshot.
>
> 2) More recent learnable diffusion models, which we compare against in our experiments, also learn from time-evolving cascades and can predict infection times or marginal infection states. As discussed in our paper, these models do not reconstruct explicit propagation edges. They describe how diffusion unfolds over discrete time but do not output a propagation tree for a single observed infection state.
>
> DIPT addresses a different observation regime. It reconstructs the propagation tree for a single infection snapshot on a given graph, without timestamps or diffusion-model assumptions, using a learned influence function and a discrete–continuous optimization procedure.
>
> > Q1: How to sample the node set s from P(s)? What is the complexity?
>
> The seed set is modeled using the latent-variable approach described in Section 3.5. During training, the model samples a latent vector from the encoder conditioned on the ground-truth seed set, and the decoder converts this latent into node-wise Bernoulli probabilities from which the seed configuration is sampled. This follows the standard VAE training procedure.
>
> During inference, we initialize a latent vector empirically as average over the training set and directly optimize it to match the observed final infection state (Section 3.7, Eq.15) . Once the optimized latent is obtained, a single decoder pass yields the seed probabilities, from which the seed set is produced.
>
> In all cases, generating a seed set requires only one decoder forward pass and node-wise Bernoulli sampling, resulting in linear $O(|V|)$ complexity.

---

### Official Review · Reviewer_VK8b · 2025-10-30

**Soundness:** 2
**Presentation:** 2
**Contribution:** 2
**Rating:** 2
**Confidence:** 3

**Summary:**

This paper claims that existing study on information propagation does not recover the actual propagation tree and then proposes a probabilistic framework DIPT to infer the propagation trees.

**Strengths:**

1.	This paper studies an important and heavily investigated research problem, which is critical in real-world applications.

2.	The illustrations are well-designed and helpful in content comprehension.

**Weaknesses:**

1.	The authors claim existing methods do not recover the “who-infected-whom” paths. However, this is a typical problem in network inference, which has been widely studied, for example, NetInf [1], NetRate [2], NMF [3], and FIM [4]. As far as I know, NMF does not assume any diffusion model. Please further justify this claim.

[1] Inferring networks of diffusion and influence. KDD, 2010

[2] Uncovering the Temporal Dynamics of Diffusion Networks. ICML, 2011

[3] Network Diffusions via Neural Mean-Field Dynamics. NeurIPS, 2018

[4] Scalable Continuous-time Diffusion Framework for Network Inference and Influence Estimation. WWW, 2024

2.	The idea of maximizing a posterior has been utilized early, e.g., NetInf and NetRate. Please justify the fundamental distinction either in the theoretical perspective or the technique perspective for better understanding.

3.	Important related work is not discussed in the related work, and related baselines are not compared in the experiment. Please refer to W1 for examples.

**Questions:**

Please refer to my weakness comments.

---

> ### Author Response · Authors · 2025-11-28
>
> We’d like to thank the reviewer for your time and effort in reviewing our paper. We address your raised points as follows.
>
> > W1: The authors claim existing methods do not recover the “who-infected-whom” paths. However, this is a typical problem in network ...
>
> The key distinction is that NetInf, NetRate, NMF, and FIM aim to learn the global diffusion network: given many timestamped infection trajectories, they estimate which edges exist and how strongly one node influences another. Their outputs are network parameters (edge weights, rates, or marginal infection dynamics), not the infection parent of each node in a specific observation.
> Unlike DIPT, they do not take “a set of infected nodes” as input and determine which node infected which in that specific outcome on a given graph.
>
> NMF, specifically, models global marginal infection probabilities using mean-field dynamics. It does not return parent assignments or event-specific propagation structures. More advanced neural diffusion models like DDMSL (which we already compare against in our model) also operate in this timestamp-supervised regime and similarly do not produce infection trees.
>
> DIPT solves a different problem: given only infection state on a graph without timestamps, without time-series trajectories, and without assuming a particular diffusion model, it infer the who-infected-whom tree for that specific observation. Prior works neither operate under this observational setting nor output such instance-level propagation trees.
>
> > W2: The idea of maximizing a posterior has been utilized early, e.g., NetInf and NetRate. Please justify the fundamental distinction either in the theoretical perspective or the technique perspective for better understanding.
>
> NetInf and NetRate also apply MAP principles, they optimize a different posterior over a different latent variable. These methods maximize likelihood with respect to global diffusion parameters such as edge existence in NetInf and continuous-time rates $\theta = \{\alpha_{uv}\}$ in NetRate, i.e. $\theta^* = \arg\max_{\theta} P(\text{temporal trajectories} \mid \theta)$. Thus, their objective is global parameter estimation under fixed generative assumptions, not recovering who-infected-whom for a specific observation.
>
> In contrast, DIPT performs MAP directly over the propagation tree $T$ for a single final infection state. Instead of assuming exponential transmission models, DIPT learns a parametric neural influence function $f_\psi(u \to v)$. The inference procedure alternates between a continuous influence-propagation update and a discrete propagation tree optimization step.
>
> Technically NetInf/NetRate use fixed diffusion likelihoods with purely continuous parameter optimization and no propagation structure/tree decoding. DIPT uses a **learnable neural influence model with continuous influence propagation + discrete parent selection**, a discrete–continuous optimization inference framework.
>
> > W3: Important related work is not discussed in the related work, and related baselines are not compared in the experiment. Please refer to W1 for examples.
>
> NetInf and NetRate are in fact already cited and discussed in our Related Work section as pioneer approaches for learning global diffusion networks from timestamped infection trajectories. We also compare against more recent and advanced formulations e.g., DDMSL and DITTO, which represent the current state of neural diffusion modeling.
>
> However, NetInf, NetRate, NMF, and FIM are not directly comparable baselines for our task, because they solve a fundamentally different inference problem. These methods learn global edge parameters or marginal diffusion dynamics under fixed generative mechanisms and require timestamped trajectories. Most of these techniques are purely likelihood-based (e.g., NetInf/NetRate optimize fixed exponential models), rather than learnable neural diffusion models.
>
> In contrast, DIPT reconstructs propagation trees for a single final infection state, using a learnable neural influence model and a discrete–continuous optimization framework. Because these classical methods do not operate under our observation regime, we instead compare against recent learnable inverse diffusion models that produce infection probabilities and/or discrete structures, such as DDMSL and DITTO, which are more appropriate and recent baselines for our setting.

---

### Official Review · Reviewer_2pnY · 2025-10-31

**Soundness:** 3
**Presentation:** 3
**Contribution:** 2
**Rating:** 2
**Confidence:** 4

**Summary:**

This paper introduces DIPT (Deep Identification of Propagation Trees), a probabilistic framework aiming to infer “who-infected-whom” propagation trees in graph diffusion processes from a single final snapshot of infected nodes, without requiring knowledge of the diffusion mechanism. The method jointly learns a diffusion model and reconstructs propagation structures via an alternating discrete–continuous optimization scheme. A variational prior is used to model source node distributions, and an iterative propagation process learns edge-wise influence strengths.

**Strengths:**

The problem formulation of propagation tree identification is novel and relevant, bridging the gap between source localization and full diffusion reconstruction.

The probabilistic treatment and alternating optimization are theoretically justified and well-detailed.

DIPT achieves strong results on multiple benchmarks, particularly in sparse and near-tree diffusion settings.

**Weaknesses:**

As acknowledged in the results, DIPT struggles on networks with dense connectivity or multiple overlapping cascades, where “who-infected-whom” relations are highly ambiguous. The model’s design (monotonic tree updates) inherently favors sparse, acyclic diffusion patterns.

The learned source prior and node features play a critical role; when these correlations weaken (e.g., random or noisy features), accuracy drops substantially (Table 9).

The datasets where DIPT excels often have clear structural constraints or ground-truth trees closely aligned with static network topology, which might inflate apparent improvements.

Although time complexity is analyzed, GPU runtime for very large graphs (e.g., Twitter-scale) may still be prohibitive.

The model outputs a single best tree, without quantifying uncertainty or providing multiple plausible propagation hypotheses.

**Questions:**

How sensitive is DIPT’s accuracy to the assumed number of propagation steps K? Would over- or under-estimating K lead to systematic errors?

Could the authors provide results or discussion on very dense diffusion graphs (e.g., >0.1 edge density), where the acyclicity assumption may be unrealistic?

Other questions please refer to Weaknesses above.

---

> ### Author Response · Authors · 2025-11-28
> **Addressing Weaknesses**
>
> We’d like to thank the reviewer for acknowledging the theoretical justification of our proposed approach. We address your raised points as follows.
>
> > W1: As acknowledged in the results, DIPT struggles on networks with dense connectivity or multiple overlapping cascades, where “who-infected-whom” relations are ...
>
> Highly dense graphs are actually the settings where DIPT shows the largest performance gains over the baselines. For example, on Memetracker the path precision is 48% higher than the best-performing baseline, and on IDSS the improvement is 30%. While dense graphs are indeed challenging for all methods, DIPT achieves higher margins here compared to less dense graphs such as Cora-ML (≈20%). We appreciate the reviewer’s observation, and clarify that the model’s design enforces acyclicity, an inherent property of propagation trees, but does not favor sparse diffusion patterns.
> The objective in Eqs. (3)–(6) maximizes
> $
> P_\psi(y \mid s, T, G)\cdot P_\phi(s),
> $
> where the size of the propagation tree (|T|) is fixed by the number of infected nodes (each non-source node must select exactly one parent). No term in the objective penalizes node degree or the number of candidate neighbors.
> During inference, the monotone update in Eq. (9) computes
>
> $
> P_{inf}^{(k)}(v) = max( P_{inf}^{(k-1)}(v),  max_{u ∈ C(v)} P_{inf}^{(k-1)}(u) · I_{uv} ),
> $
>
> and the parent selection in Eq. (10) chooses
>
> $
> p_v = \arg\max_{u \in C(v)} P_{inf}^{(K)}(u)\cdot I_{uv}.
> $
>
> Both steps evaluate all infected neighbors `C(v)` without imposing any penalty on large neighborhoods. Thus, increasing graph density only enlarges the candidate set.
>
> > W2: The learned source prior and node features play a critical role; when these correlations weaken (e.g., random or noisy features) ...
>
> This supports one of the core motivations and component of the paper that features influence propagation significantly(Line 067-069) and motivates DIPT design: DIPT integrates both feature-based influence and structural information when inferring propagation. Table 9 also supports and shows node features' importance. However, even under substantial corruption (10 dB noise), DIPT still achieves ~43% path precision, indicating that structural information alone remains also informative and optimization framework shows robustness.
>
> > W3: The datasets where DIPT excels often have clear structural constraints or ground-truth trees closely aligned ...
>
> This concern is closely related to W1. Ground-truth trees may align more closely with topology in less dense graphs, but this does not inflate our results. The largest performance gains occur on denser and structurally ambiguous datasets such as Memetracker, and IDSS, where parent identification is substantially harder. Compared to baselines, DIPT achieves higher margins on these challenging datasets than on less dense graphs like Cora-ML.
>
> > W4: Although time complexity is analyzed, GPU runtime for very large graphs (e.g., Twitter-scale) may still be prohibitive.
>
> Compared to prior approaches, DIPT scales more favorably. Methods such as DDMSL and DDMIX rely on reconstructing or simulating discrete infection timestamps, causing their runtime to grow with the number of diffusion steps T in addition to graph size. In contrast, DIPT does not depend on fine-grained timestamps: inference consists of a fixed number of propagation passes K, each operating over the edges with cost proportional to |E|. Since empirically K is held constant across datasets, DIPT avoids the additional dependence on T and exhibits more predictable scaling. As shown in the appendix, DIPT remains competitive on the largest graphs in our benchmark.
>
> > W5: The model outputs a single best tree, without quantifying uncertainty or providing multiple plausible propagation hypotheses.
>
> The “single best tree” output reflects the decoding objective of the task, where the goal is to recover the most likely propagation tree. DIPT relies on probabilistic edge influence scores during inference, the probabilistic formulation makes extensions such as top-k decoding or sampling-based variants straightforward, but outside scope of our work presently. During early training iterations we already use a small beam search to stabilize optimization (Appendix B) to infer three most probable paths to each infected node instead of only one.

---

> ### Author Response · Authors · 2025-11-28
> **Addressing Questions**
>
> > Q1: How sensitive is DIPT’s accuracy to the assumed number of propagation steps K? Would over- or under-estimating K lead to systematic errors?
>
> In DIPT, K only controls how many iterations of influence propagation we run. The update
>
> $P_{\text{inf}}^{(k+1)}(v) = \max\left( P_{\text{inf}}^{(k)}(v),\; \max_{u\in C(v)} P_{\text{inf}}^{(k)}(u)\cdot I_{uv} \right)$
>
> is monotone and convergent, so there exists a small $ K^* $ after which the influence scores stop changing. For any $K \ge K^{*}$, the parent selection is similar, and the inferred propagation tree is fully stable.
>
> If K is set significantly below this convergence depth, the model has under-propagated influence: long-range information has not fully reached all nodes. In this case, some nodes may rely more on local influence rather than the fully converged global scores,  but it is not systematically erroneous. Empirically, convergence occurs well before $K = 100$, so using a fixed $K = 100$ ensures stable trees across all datasets. We evaluate path precision across different K steps on Cora-ML from 30 to 70 with interval of 10:
>
> |(K)          | 30 | 40 | 50 | 60 | 70 |
> |-----------------|--------|--------|--------|--------|--------|
> | Path Precision  | 0.587  | 0.619  | 0.618  | 0.622  | 0.622  |
>
>
> > Q2: Could the authors provide results or discussion on very dense diffusion graphs (e.g., >0.1 edge density), where the acyclicity assumption may be unrealistic?
>
> Acyclicity is a property of the propagation tree, not of the underlying graph. In our problem setting, an infected node cannot reinfect its infector, and each node has exactly one infection source; therefore, the realized cascade is always a directed tree, regardless of how dense the underlying network may be. Dense graphs increase the number of plausible parents, but they do not introduce cycles into the diffusion sequence. We study single-source infection chains without reinfection loops, so the propagation structure must be acyclic.
>
> The IDSS mobility network used in our experiments is highly connected, with an observed edge density of 0.073, which falls well within the dense-graph regimes. DIPT performs strongly on this dataset(30% better than best performing baselines), showing that high underlying connectivity increases candidate-parent complexity but does not conflict with the acyclic diffusion model or the correctness of the reconstructed propagation tree.

---

### Official Review · Reviewer_F39n · 2025-10-31

**Soundness:** 4
**Presentation:** 3
**Contribution:** 3
**Rating:** 6
**Confidence:** 3

**Summary:**

This paper introduces DIPT, a novel probabilistic framework for tackling the challenging inverse problem of propagation tree identification. Given only a single, final snapshot of infected nodes in a graph, along with node features, DIPT aims to reconstruct the entire propagation tree. The core of the method is a discrete-continuous alternating optimization strategy that jointly learns an unknown, feature-based diffusion mechanism and infers the latent tree structure without requiring any direct supervision on propagation paths. The method's effectiveness is demonstrated through extensive experiments on eight diverse datasets.

**Strengths:**

The paper formally defines and addresses "propagation tree identification from a single snapshot," a problem that is substantially more challenging and informative than the well-studied task of source localization. The proposed DIPT framework is technically sound and novel, creatively integrating a learnable influence model, a variational prior for sources, and an alternating optimization scheme into a coherent system. The paper is written with good clarity, making the complex methodology easy to follow. Finally, the empirical evaluation is of good quality, demonstrating state-of-the-art performance across multiple real-world and synthetic datasets against strong baselines.

**Weaknesses:**

- The model's strong and unexamined dependency on the availability and quality of node features. In the setting of some other works, the feature of the nodes is usually unknown. However, the core influence function, `f_ψ(F_u, F_v)`, is entirely feature-driven. This potentially limits the method's applicability in domains where informative node features are scarce or unavailable (e.g., anonymous networks). The experiments, while extensive, do not sufficiently probe this limitation. For instance, an evaluation and comparison on featureless graphs is missing, making it difficult to disentangle the contribution of the novel optimization framework from the predictive power of the input features themselves.

**Questions:**

1.  The methodology critically assumes the input is a "final" snapshot of the diffusion process. In real-world applications, how can one reliably determine if an observed state is final? Could you comment on the robustness of DIPT and the validity of the inferred tree if the input is actually an **intermediate snapshot** where diffusion is still ongoing?
2.  Given the model's reliance on node features, what would be its performance on graphs where only structural information and infection indicators are available? A discussion on this would help clarify the method's practical application boundaries.

---

> ### Author Response · Authors · 2025-11-28
>
> We’d like to thank the reviewer for acknowledging the novelty and importance of the studied problem. We address your raised points as follows.
>
> > W1 and Q2: The model's strong and unexamined dependency on the availability and quality of node ...
>
> We appreciate the reviewer’s observation, and clarify that DIPT is not completely feature-only dependency. The model incorporates structural information through the propagation updates and the GNN layers for encoding features(described in Appendix B), so $f_{\psi}$ operates on representations that has both node features and structural information as well. This is also reflected in Table 9: when node features are heavily perturbed, DIPT still achieves around 52% path precision, which shows that the structural component alone also provides informative predictive information and optimization framework shows robustness. The ablation therefore directly probes the mentioned concern. To further evaluate the structural only information, we also report a setting where we use only structural information: positional embeddings, Laplacian embeddings.
>
> | Setting                       | Path Precision (CORA-ML) |
> |------------------------------|----------------|
> | Baseline (True Features)     | 0.622          |
> | Gaussian Noise (SNR = 20 dB) | 0.528          |
> | Gaussian Noise (SNR = 10 dB) | 0.433          |
> | No Features(Laplacian Positional Embeddings used) | 0.411          |
>
> While node features naturally contribute in real world diffusion settings, the results show that structural information by itself also carries informative signals, and DIPT can successfully incorporate it and optimization framework can be applicable and effective in anonymous graphs as well.
>
> > Q1: The methodology critically assumes the input is a "final" snapshot of the diffusion process. In real-world ...
>
> DIPT operates on an observed infection state and reconstructs the most likely propagation tree. While our evaluation provide final infection snapshots, the inference procedure does not inherently rely on finality. If the observed state is intermediate, DIPT simply produces the partial propagation tree among the nodes that are already infected at the observation time.
>
> To concretely assess robustness, we evaluated this on the Twitter cascade dataset. Instead of providing the full set of infected nodes (the final snapshot), we constructed an intermediate snapshot by taking only the first half of the infected nodes in the temporal order of the cascade. We then inferred a propagation tree from this intermediate state and compared it with the tree inferred from the full final state, restricted to the nodes that appear in both snapshots.
>
> We found that 92.3% of their parent–child edges were identical, and path precision for intermediate path is 58.8%(for final state it is 58.1%). Only a small fraction changed due to later infections altering the global likelihood. This confirms that algorithm is stable with respect to intermediate observations and that its core structure extraction remains accurate even when the diffusion process is not fully complete.

---

### Author Response · Authors · 2025-12-04

We sincerely thank all the reviewers for their valuable reviews. We also deeply appreciate the AC’s time in reviewing this rebuttal, especially given the additional workload for this year's ICLR.

Unfortunately, no reviewer was able to respond during the short window between the November 27 data leak and the rebuttal posted. However, we would like to summarize the reviews and rebuttal in response to reviewers.

**Strengths highlighted by reviewers**

* Both Reviewer F39n and Reviewer 2pnY noted the novel formulation of the propagation-tree identification problem, emphasizing that DIPT tackles a substantially harder and previously unaddressed setting compared to prior works.
* Reviewer 2pnY also acknowledged the theoretical justification of the proposed method. Reviewer VK8b and Reviewer HTNY appreciated the relevance of the problem and the usefulness of the illustrations for clarifying the methodology

**Concerns and responses**

* Feature dependence (F39n, 2pnY):
  We clarified that DIPT is not only dependent on node features and integrates structural information as well. New ablations with missing features showed consistent robustness and strong performance using only structural information.

* Intermediate snapshots (F39n):
  We showed in the rebuttal that DIPT remains stable when the observed infection state is not final, because primarily DIPT reconstructs the path given the observed state. On Twitter cascades, 92% parent–child edges matched the final inference, showing near-identical path structure.

* Performance on Dense or ambiguous graphs (2pnY):
  We clarified that DIPT actually performs best on dense datasets such as Memetracker and IDSS, with the largest gains over baselines. We also clarified in response to reviewer that acyclicity is a property of propagation trees, independent of graph density.

* Relation to NetInf, NetRate, NMF, FIM and baselines (VK8b):
  We explained that these methods infer global diffusion parameters from many timestamped cascades and do not recover instance-level who-infected-whom trees from a single snapshot without timestamps, which is the central task addressed by DIPT. Two of the mentioned  works by reviewer (NetInf and NetRate) have already been explicitly discussed and cited in the paper, for the other mentioned paper much advance and recent baselines like DDMSL and DITTO have been already been compared as baselines.

* MAP distinction and baselines (VK8b):
  We clarified that previous MAP approaches optimize global model parameters, whereas DIPT performs MAP over the propagation tree itself using a learned neural influence function and discrete–continuous inference.

* Sensitivity to K and seed sampling (2pnY, HTNY):
  We provided results showing stable performance with respect to K hyperparameter. Also in reply to reviewer HTNY, we clarified that seed sampling follows a standard VAE procedure and has linear complexity.

---

### Note · Authors · 2026-01-20

I have read and agree with the venue's withdrawal policy on behalf of myself and my co-authors.